# Effects of Brines and Containers on Flavor Production of Chinese Pickled Chili Pepper (*Capsicum frutescens* L.) during Natural Fermentation

**DOI:** 10.3390/foods12010101

**Published:** 2022-12-25

**Authors:** Shiyao Zhang, Yue Xiao, Yongli Jiang, Tao Wang, Shengbao Cai, Xiaosong Hu, Junjie Yi

**Affiliations:** 1Faculty of Food Science and Engineering, Kunming University of Science and Technology, Kunming 650500, China; 2Yunnan Engineering Research Center for Fruit & Vegetable Products, Kunming 650500, China; 3College of Food Science and Nutritional Engineering, China Agricultural University, Beijing 100083, China

**Keywords:** fermented chili pepper, flavor, brines, containers, multivariate data analysis

## Abstract

The effects of (fresh/aged) brine and (pool/jar) containers on the flavor characteristics of pickled chili peppers were investigated based on a multivariate analysis integrated with kinetics modeling. The results showed that the effect of brine on organic acid, sugar, and aroma was more dominant than that of containers, while free amino acids production was more affected by containers than brines. Chili pepper fermented using aged brine exhibited higher acidity (3.71–3.92) and sugar (7.92–8.51 mg/g) than that using fresh brine (respective 3.79–3.96; 6.50–9.25 mg/g). Besides, chili peppers fermented using pool containers showed higher free amino acids content (424.74–478.82 mg/100 g) than using a jar (128.77–242.90 mg/100 g), particularly with aged brine. As for aroma, the number of volatiles in aged brine was higher (88–96) than that in fresh brine (76–80). The contents of the esters, alcohols, and ketones were significantly higher in the aged brine samples than those in fresh brine (*p* < 0.05), while terpenes in chili pepper fermented using the pool were higher than those using the jar. In general, jar fermentation with aged brine contributed more flavor to pickled chili peppers than other procedures.

## 1. Introduction

Natural fermentation is an ancient technique widely used for food processing. It can extend shelf life, improve flavor profile, and increase nutritional and functional values [1,2]. Among these, the flavor characteristic is one of the main indicators directly affecting consumers ’preferences for fermented vegetable products. The flavor properties of final fermented vegetable products have been proven to be closely related to raw materials, processing methods, and environmental conditions [3]. The production of fermented vegetates depends on the spontaneous fermentation of lactic acid bacteria (LAB) and yeast in the raw materials and brine [4,5]. In addition to the raw material, the fermentation brine is the main controlled variable to stable quality properties for Chinese traditional pricked vegetable processing. For industrial production, two types of procedures for fermented brines are commonly applied: (1) using all fresh brine and (2) using fresh brine mixed with reused brine at a certain ratio [4,6]. Recycling and using aged brine as a starter for fermentation can not only be friendly to the environment, but can also be valuable for unique flavor formation [7]. However, to date, little information has been reported regarding the role of fresh and aged brine on the flavor production of pickled vegetables during fermentation.

Pickle jars are traditionally used as the main fermenting vessels to provide the fermented vegetable with a unique flavor, particularly porcelain jars [8]. Anaerobic or/and microaerobic spontaneous fermentation occurs when using pickle jars because they need to be sealed by adding water to the moat, not only to prevent air from entering but also to allow the carbon dioxide produced by fermentation to escape [3]. Considering the limited capacity of the pickle jar, large-scale fermentation pools have been widely applied in the industrial production of fermented vegetables. The fermented vegetables and brine are placed in the pool, flattened, covered with two layers of dense plastic nets, and tightly pressed using wide wooden boards. The brine is periodically circulated by the pump for aerobic natural fermentation [7,9]. Even though our previous study found that vegetables fermented using small-scale household pickle jars seem to contain additional various aroma compounds than those fermented using large-scale industrial pickle pools [3], it remains unclear whether the different fermentation containers in industrial production, e.g., large-scale fermentation jars and pools, play any direct role in the flavor formation of fermented vegetables.

Pickled chili pepper is one of the most typical fermented vegetables in southwest China [6]. Our research team has carried out some studies on the quality characteristics of Chinese traditional fermented chili peppers [1,3,4,5,10,11]. It was found that fermentation contributed to chili peppers’ taste-related attributes, such as increased organic acids contents and free amino acids (FAAs) contents [4]. In addition, esters (4-methylpentyl 2-methylbutanoate, 4-methylpentyl 3-methylbutanoate, methyl salicylate, 3-methylbutyl 2-methylbutanoate, pentyl 3-methylbutanoate) and terpenes (*β*-myrcene, linalool, (E)-linalool oxide, neryl oxide, and α-terpineol) were the main aroma compounds detected in fermented chili peppers [11]. During fermentation, some aromatic volatile compounds clearly increased in pickled chili peppers, mainly those including terpenes, alcohols, and aldehydes [4]. In addition, it was found that aged brine fermentation promoted the growth of LAB and yeast. This led to the rapid consumption of reducing sugar, inhibited undesirable enterobacteria, and reduced the production of nitrite and biogenic amines during the fermentation of *Paocai* [12]. The plastic jar brine samples contained higher levels of lactic acid and threonine, while more abundant volatile compounds were evident in the porcelain jar [13].

Based on these previous studies, it can be deduced that fermentation brines and containers are the main variables that the industry has paid high attention to and wants to control in order to produce high quality products with stable flavor properties. Therefore, it is necessary to systematically study the effects of fermentation brines and containers on the quality of fermentation products, in order to obtain the high-quality fermentation methods in a large-scale industrial production. However, such research is obviously lacking at present. To illustrate this issue, the present study aimed to evaluate the role of different brines (fresh/aged brine) and large-scale containers (pool/jar) on the flavor formation of fermented chili peppers during natural fermentation. Taste-related attributes (organic acid, FAA, and sugar) and aroma-related attributes (headspace volatile compounds) of chili pepper were investigated during the fermentation. The results of this study could guide the industry to select proper fermentation procedures (brines and containers) for different types of fermented chili peppers production.

## 2. Materials and Methods

### 2.1. Chemicals and Reagents

Standards (oxalic, tartaric, quininic, malic, lactic, acetic, citric, succinic, fumaric acids, fructose, glucose, and 3-octanol) were purchased from Aladdin (Shanghai, China). High-performance liquid chromatography (HPLC)-grade acetonitrile and methanol were obtained from Sigma-Aldrich (Darmstadt, Hessen, Germany). The standard amino acid solution (type H) was acquired from Wako (Wako-shi, Japan). *N*-Alkanes (C_5_–C_10_, C_10_–C_25_) were provided by Anpel (Shanghai, China).

### 2.2. Sample Preparation and Collection

Fermented chili pepper samples were produced by Yunnan Hongbin Green Food Group Co., Ltd. (Jianshui, Yunnan, China) under large-scale fermentation procedures including pickle pool with fresh brine (PFB), pickle pool with aged brine (PAB), pickle jar with fresh brine (JFB), and pickle jar with aged brine (JAB). Fresh green *Xiaomila* (*Capsicum frutescens* L.) were fermented in pools or jars with fresh brine (containing 8.16% sodium chloride, 0.22% calcium chloride, 0.63% glacial acetic acid, and water) or aged brine (fresh brine: reused brine, 1:4) at an ambient temperature (20–25 °C) for a month. In the preliminary experiment, the changes in pH and OD_600 nm_ of the fermented chili peppers followed by different fermentation procedures were monitored (data not shown). Based on the results of the preliminary experiment, the sampling time moments were selected as 0, 1, 4, 7, 14, and 30 days during fermentation. For each sampling moment and fermentation procedure, samples (~100 g) were taken at three sampling points from the top to the bottom of the pool/jar, and four sampling points from the side to the center of the pool/jar. The collected samples for each time points (~700 g) were mixed into sterile bags. Sterile bags were used for sampling to reduce the external effects on microorganisms. After sampling, they were quickly frozen using liquid nitrogen and immediately transported to the laboratory for physicochemical parameters and flavor analysis.

### 2.3. pH Value

The pH value was determined according to the previous study [3]. Briefly, 10 g of fermented chili peppers were homogenized and vortexed with 90 mL deionized water. After filtering, the pH value of the filtrate was measured using a pH meter (Lei-ci, Shanghai, China). The measurements were carried out in triplicate.

### 2.4. Organic Acids

The extraction and measurement of organic acids in the fermented chili peppers samples were conducted as per our previous study [4]. Briefly, the organic acid concentration was analyzed using a HPLC (Agilent 1200, Agilent Technologies, Santa Clara, CA, USA) equipped with a Prevail Organic Acid column (250 mm × 4.6 mm, 5 μm, Avantor, Radnor, PA, USA) and a UV detector set at 210 nm. The mobile phase (25 mmol/L potassium dihydrogen phosphate buffer, pH 2.5) was used at a flow rate of 0.8 mL/min. The injection volume was 30 μL. Organic acid standards were used as an external standard for the identification and quantification of organic acids including oxalic, tartaric, quininic, malic, lactic, acetic, citric, succinic, and fumaric acids. Each sample was extracted and analyzed in triplicate.

### 2.5. Sugars

The extraction of sugar in the fermented chili peppers samples was the same as for the organic acids. The extract was analyzed by an HPLC system (Agilent 1260, Agilent Technologies, Santa Clara, CA, USA) and an ELSD detector (1260 Series, Agilent Technologies, Santa Clara, CA, USA) equipped with a column (4.6 mm × 250 mm, 5 μm, Asahipak NH2P-50 4E, Shodex, Japan). The analysis condition was set at 1 mL/min with an injection volume of 5 μL, and isocratic elution with water and acetonitrile (25:75, *v*/*v*). Glucose, fructose, and sucrose were identified and quantified with their respective standards. Each sample was extracted and analyzed in triplicate.

### 2.6. Free Amino Acids

FAAs were extracted and analyzed as per our previous study [11]. The homogenate of fermented chili peppers (0.5 g) was vortexed with trichloroacetic acid (10 g/L) and placed at room temperature for 1 h. Then, the mixture was centrifuged at 4000× *g* for 20 min and filter supernatant with 0.45 μm syringe filters. The amino acid analyzer (L-8900, Hitachi, Tokyo, Japan) was equipped with an ion-exchange resin 2622 column (4.6 mm × 60 mm, 3 μm) and a UV detector was used at 440 (proline) and 570 (other FAAs) nm. Amino acid mixture (type H) was used as the standard to calculate the FAA content. Each sample was extracted and analyzed in triplicate.

### 2.7. Volatile Compounds

The volatile compounds of the samples were determined by using gas chromatography-mass spectrometry (GC-MS) (QP2010, Shimadzu, Kyoto, Japan) coupled with headspace-solid phase microextraction (HS-SPME), as described in our previous study with some modifications [5]. Briefly, 3 g homogenous sample and 3 mL saturated NaCl solution were mixed in a headspace vial with 100 μL 3-octanol as the internal standard. After incubating at 40 °C for 15 min under agitation at 500 rpm, the SPME fiber coated with 50/30 μm (Zhenzheng, Qingdao, China) was used to extract the volatile compounds under the same condition for 40 min. Then, the volatile compounds were thermally (250 °C) desorbed from the fiber into the injector port of the GC for 5 min.

A GC-MS system fitted with a DB-5 MS column (30 m × 0.25 mm × 0.25 μm, Agilent Technologies, Santa Clara, CA, USA) was used to separate and detect the volatiles. The initial temperature of the GC column oven was kept at 45 °C for 5 min, then increased to 250 °C at a rate of 5 °C/min, maintained for 2 min, and then cooled to the initial temperature. The carrier gas was helium at a flow rate of 0.74 mL/min. The scanning range of the mass spectrometer was *m/z* 35–500, the electron ionization mode was 70 eV, and the ion source temperature and interface temperature were 230 °C and 280 °C, respectively. Each sample was subjected to three parallel experiments.

Tentative identification of volatile compounds was performed based on the comparison of the database from the NIST 2014 library and the experimentally determined retention index (RI), which was calculated using *n*-alkanes (C_5_–C_10_, C_10_–C_25_, Anpel, Shanghai, China) under the same operating conditions. Internal standard (3-octanol) calibration was also conducted for semi-quantification.

### 2.8. Data Analysis

#### 2.8.1. Statistical Analysis

Data were presented as mean value ± standard deviation. Results were analyzed using one-way analysis of variance (ANOVA) and the Origin 2021 software (Origin Lab Corporation, Northhampton, MA, USA). Tukey’s test using IBM SPSS (version 26.0, IBM Corp., Armonk, NY, USA) was performed at a significance level of 0.05.

#### 2.8.2. Multivariate Data Analysis (MVDA)

The clustered heatmap was illustrated for organic acids and FAAs data by TBtools (version 1.098, CJ-Chen, China). The influence of brines and containers on flavor compounds was analyzed using a partial least squares discriminant analysis (PSL-DA) model on Solo (Version 9.1, Eigenvector Research, Manson, WA, USA). The flavor compounds were considered as *X* variables, and brines and containers were considered as categorical *Y* variables. Variable identification (VID) coefficients were calculated to select discriminant aroma compounds, and those with absolute VID values above 0.800 were selected [5].

#### 2.8.3. Kinetic Modeling

Fractional conversion kinetic modeling was carried out by using Equation (1) in Origin (Version 2021, Origin Lab Corporation, Northhampton, MA, USA) to fit the pH changes of 4 fermented chili peppers with the fermentation time.
(1)$$C=C_{\infty }+\left(C_{0}-C_{\infty }\right)exp\left(kt\right)\;$$
where *C* is the pH at fermentation time (days), *C*_0_ is the initial value on day 0 of fermentation, *C_∞_* is the value of the stable fraction, *k* is the reaction rate constant, and *t* is the number of days of fermentation.

## 3. Results and Discussion

### 3.1. Taste Properties

#### 3.1.1. pH Value

The pH value is the main parameter for determining the fermentation stages of pickled vegetables [14]. As shown in Figure 1A, the changes of pH value in four fermented chili peppers were statistically significant during fermentation (*p* < 0.05). The pH of fermented chili peppers significantly decreased within 7 days (*p* < 0.05) and then maintained a stable value during fermentation. After 30 days, the pH values of fermented chili peppers varied from 3.71 and 3.96 (PFB: 3.96 ± 0.02, PAB: 3.92 ± 0.02, JFB: 3.79 ± 0.00, JAB: 3.71 ± 0.01) and fell within the ranges of 3.2–4.2, indicating that the peppers in this study were considered to be fully fermented and ready to eat [3]. During fermentation, LAB metabolizes carbohydrates, resulting in a rapid drop in pH [15].

Under the same fermentation container, the pH value of chili peppers fermented with aged brine was lower than that fermented with fresh brine (Figure 1A). The initial loading of microorganisms might contribute to a vigorous metabolism, producing more organic acid and resulting in a lower pH [12]. The evolution of pH as a function of fermentation time was well modeled using a first-order fractional conversion model (*R*^2^ > 0.90), and the change rates of JFB (*k* = 0.62) and JAB (*k* = 0.69) were higher than that of PFB (*k* = 0.39) and PAB (*k* = 0.41) (Figure 1A). This may be due to the fact that the jar was a closed environment compared to an open fermentation pool, where the metabolism of hetero-fermentative microorganisms produced carbon dioxide (CO_2_) [4,16]. An increase in CO_2_ leads to an acidification cultivation broth, which leads to an increase in the pH of the fermented chili peppers [17].

#### 3.1.2. Organic Acids Profile

The contents of organic acids play important roles in the sour taste of pickled vegetables. In this study, a total of 8 organic acids (except for fumaric acid) were detected in fermented chili peppers. The changes in organic acid content during fermentation of four kinds of peppers are shown in Figure 1B, which can be divided into three categories. Category B_1_ included most of the aged brine group fermented during 4–30 days, category B_2_ included the fresh brine group fermented during 4–30 days, and category B_3_ included all chili pepper samples fermented during 0–1 days. It indicated that the effects of fermentation brines were dominant in that of containers on the organic acid profile. In addition, there were large gaps in the organic acid profile between pickled chili peppers fermented before and after 4 days (Figure 1B). To zoom in on the data, the organic acids content of fermented chili peppers in aged brine was significantly higher than that in fresh brine (*p* < 0.05) (Figure 1B). Aged brine was enriched in LAB, which could produce more organic acids compared to fresh brines [18]. Besides, the content of organic acids increased with the fermentation time, reached the highest at 14 days, and then decreased (Figure 1B). This might be explained by the fact that organic acids can serve as carbon sources for microbial growth [5]. In addition, the LAB could efficiently metabolize organic acids to other flavor compounds, such as esters, ketones, alcohols, and aldehydes [19]. Compared with PFB, JFB, and JAB, PAB had the lowest acetic acid content after 30 days of fermentation (Figure 1B), which may be due to the formation of esters combined with alcohols, thus contributing to the sourness and aroma of fermented chili peppers [19]. Among all samples, JAB samples had the highest total organic acids content, corresponding to the pH results (Figure 1A).

#### 3.1.3. Sugars Profile

Figure 2 showed the change of sugar content in different fermented chili pepper samples during 30 days of fermentation. Glucose and fructose were detected in fermented chili peppers, whereas sucrose was not detected. Similar results were also found in a previous study, where non-reducing sugars were not detected in brine-pickled sauerkraut [20]. It was reported that reducing sugar in vegetables was the major carbon source for microorganisms in *Paocai* during fermentation [12]. Both glucose and fructose contents showed a decreased trend during fermentation (Figure 2A,B), probably because they were the main carbohydrates converted to lactic acid, which provided the taste and aroma of fermented vegetables [20]. Besides, the degree of utilization of glucose was higher than that of fructose, which was consistent with the results on sauerkraut [20].

The glucose content in PAB and JAB increased and then decreased (Figure 2A), which might be due to the abundance of microorganisms in the aged brine [21]. Many microorganisms could simultaneously use a variety of sugars as carbon sources to produce flavor substances [22]. For example, 2,3-butanediol and acetoin have obtained good fermentation yields with glucose or sucrose as a carbon source [23]. The *Lacticaseibacillus casei* and *Lactiplantibacillus plantarum* ferment lactose through the phosphoenol pyruvate phosphotransferase system to produce glucose and galactose-6-phosphate [24]. In addition, the total sugar content decreased gradually in all samples; in particular, the consumption of fresh brine was faster than that of aged brine (PFB, PAB, JFB, and JAB consumed 20.30%, 16.34%, 43.43%, and 1.72%, respectively) (Figure 2C), indicating that the sugar consumption rate depends on the type of brine used for fermentation.

#### 3.1.4. Free Amino Acids Contents

Free amino acids are one of the main flavor substances in fermented vegetables, of which quality and quantity play important effects on the flavor quality [25]. In this study, 17 FAAs were identified in four fermented chili pepper samples (Figure 3), which imparted the umami taste, bitterness, and sweetness [3]. Fermented chili peppers had the highest content of bitter amino acids, followed by sweet amino acids, and had the lowest content of umami amino acids (Appendix A).

Figure 3 showed the changes in FAA content of four kinds of chili peppers during fermentation, which can be divided into three categories. Category B_1_ was the samples without fermentation, category B_2_ was the jar-fermented samples, and category B_3_ was the pool-fermented samples. It indicated that the effect of fermentation containers on the FAAs content of fermented chili peppers was more dominant than that of fermentation brines. Compared with the pickle jar, chili peppers fermented using the large-scale pool showed higher FAAs content, particularly for samples with aged brine (Figure 3). In other words, the highest amounts of FAA contents were detected in PAB samples among the samples (PFB: 424.74 ± 2.60 mg/100 g, PAB: 478.82 ± 1.26 mg/100 g, JFB: 128.77 ± 1.72 mg/100 g, JAB: 242.90 ± 2.11 mg/100 g) (Figure 3 and Appendix A). It might be because that fermentation in the pool involved the mixed fermentation of multi-microorganisms and multi-enzyme catalysis [26].

The concentration of FAAs increased until day 4 and then decreased during fermentation (Figure 3). The changing trend was similar to other brine-pickled vegetables [27]. The increase of FAAs contents might be related to the utilization of soluble proteins by LAB (e.g., *Lactiplantibacillus plantarum*) through the secretion of peptide enzymes and the breakdown of protein into FAAs [4,11]. The reason for the decline of the FAA content might be that FAAs were essential substances for microbial metabolism [28]. It could be engaged in the energy metabolism process (ATP) and the production of other biomass molecules via transamination, degradation, and other processes [25]. In addition, some studies have proposed that LAB could overcome the low pH stress of brine-pickled vegetables by utilizing free amino acids to produce biogenic amines [29,30].

### 3.2. Aroma Properties

Representative total ion chromatograms of headspace volatile components of fermented chili peppers, using different brines and containers when fermented for 30 days are depicted in Figure 4. Even though the chromatograms of volatiles on four groups of fermented chili peppers seem to be similar, the PAB samples showed the highest numbers of volatile compounds (96), followed by the JAB (88), PFB (80), and JFB (76). Among the volatile compounds, esters were the most abundant volatile components in all fermented chili peppers, followed by terpenes (Figure 4A–D). The 4-methylpentyl 2-methylbutanoate, 4-methylpentyl 3-methylbutanoate, 4-methylpentyl 4-methylpentanoate, and methyl salicylate were the main esters detected in all fermented chili peppers, which contributed fruity and floral notes [3]. The (Z)-*β*-ocimene and limonene were the main terpenes detected in all fermented chili peppers, which contributed citrus and floral odors [31]. Comparably, the number of alkanes, alcohols, and acids was less than esters and terpenes. In addition to the common volatile compounds, some unique compounds were also detected in chili peppers fermented by different procedures. For example, methyl 4-methylpentanoate and styrene only appeared in chili peppers fermented in pools and jars, respectively. Methyl 4-methylpentanoate was mainly produced by *Bacillus amyloliquefaciens*, which imparts a sweet and pineapple aroma to fermented chili peppers [32,33]. Styrene was only found in jar-fermented chili peppers, which might be due to the action of *Debaryomyces hansenii* [34].

Considering the large dataset obtained, PLS−DA modeling was used to determine the correlation of fermentation methods with volatiles (Figure 5). During modeling, three latent variables (LVs) were selected as being optimal to describe the volatile compounds and explained in total 99% of *Y*−variance. Since each LV explained an equal proportion of the *Y*−variance, all three plots are shown. Figure 5 clearly showed that four treatment groups of fermented chili pepper presented distinct separations, indicating different fermentation brines and containers resulted in different aroma profiles of fermented chili peppers. As for the loading plots, more volatiles (shown with small open circles) were clustered around the aged brine groups than the fresh ones, indicating that aged brines were rich in aroma compounds. It was consistent with the observation in Figure 4. Similar results were reported in other studies, with a partial decrease in the flavor of freshly brined capers [35]. Traditional Chongqing radish *Paocai* fermented with aged brine was considered to have the most intense flavor and authentic taste [36].

In this study, the volatiles with an absolute VID value higher than 0.800 was selected as discriminant volatile compounds and represented by bold circles (Figure 5). Most discriminant volatiles in the jar with aged brine was associated with a positive VID value, including hexyl butanoate, 3-methylbutyl 2-methylbutanoate, hexyl 3-methylbutanoate, isopentyl hexanoate, heptyl hexanoate, amyl 2-methylbutyrate, and 4-methylpentyl isobutyrate. It indicated their contents were significantly higher in chili peppers fermented in the jar with aged brine compared to other samples. A similar observation was reported in wine fermentation; the wine fermented in barrels showed a higher concentration of alcohols and esters than that fermented in tanks [37]. In order to better understand the change trends on key aroma compounds, the discriminant volatile compounds are individually plotted in Figure 6.

As shown in Figure 6, esters were the most abundant discriminant volatiles in fermented chili peppers. Most of esters were the highest in JAB and lowest in JFB, including 4-methylpentyl isobutyrate, heptyl hexanoate, amyl 2-methylbutyrate, isopentyl isovalerate, and (Z)-3-hexenyl 3-methylbutanoate (Figure 6). These compounds had odor notes of green, fresh, fruity, and apple, as reported in the literature [4,38]. It might be because the microorganisms in fresh brine are complex and need to adapt to the environment to stabilize, resulting in fewer flavor compounds [12]. In addition, 3-methylbutyl 2-methylbutanoate only appeared in the jar with aged brine fermented chili peppers to give them a fruity aroma [11]. This might be because only the jar with aged brine contained *Pichia fermentans* and *Pichia anomala*, and *Pichia fermentans* and *Pichia anomala* were positively correlated with the content of 3-methylbutyl 2-methylbutanoate [39].

Terpenes were the second largest aroma compounds in fermented chili peppers (Figure 6). Most of them were at higher concentrations in the pool, such as *β*-chamigrene, (Z)-*β*-ocimene, *o*-cymene, methyl-1-naphthalene, and durene, imparting a floral and herbal aroma to the fermented chili peppers [10]. *Metschnikowia pulcherrima* is characterized by an extra-cellular *α*-arabinofuranosidase, which influences the content of terpenes in the fermented product [40]. *Metschnikowia pulcherrima* seems to display a high aerobic respiratory metabolism and requires high levels of oxygen [41]. In this study, the pool fermentation was an open fermentation with brine periodically circulated by a pump, while the pickle fermentation was a sealed fermentation by adding water to the moat. The oxygen content of the pool fermentation was higher than that of the jar fermentation, which seems to be more suitable for the growth of *Metschnikowia pulcherrima*. The high content of most terpenes in the pool might be related to the high level of *Metschnikowia pulcherrima*.

Most alkanes presented a significantly higher content in aged brines than in fresh brines, including heptadecane, hexadecane, octadecane, and naphthalene (*p* < 0.05) (Figure 6). Similar results were reported in previous studies. For example, many volatile compounds such as naphthalene and straight chain alkanes were produced by microbial metabolism in the second round of fermentation of radish [25]. As the alkanes threshold was high, they were not thought to directly contribute to the typical odor of chili peppers [4].

Alcohols, acids, and ketones were the other main volatile compounds representing the floral, sweet, sour, and green notes in fermented chili peppers [26]. Linalool, as the floral odor characteristic compound of *Paojiao* [3], showed significantly higher concentration in the pool than that in the jar (*p* < 0.05) (Figure 6). Our previous study found that *Companilactobacillus versmoldensis* and *Levilactobacillus brevis* were the dominant bacterial species that were positively correlated with linalool, and might be more abundant in the pool [4]. In addition to linalool, the contents of isohexanol, hexanol, acetic acid, 3,7-dimethyl-6-octenoic acid, and 3,5-dimethyl-2-octanone were also higher in the aged brine than those in the fresh brine (Figure 6). LAB and yeast seemed to be more abundant in the aged brine than the fresh brine [12]. The cell metabolism of LAB and yeast during fermentation might be related to the high content of most alcohols, acids, and ketones in the aged brine [11]. As aroma production might be highly related to microbial actions, future work will further investigate microorganism profiles among different fermentation procedures, in order to better understand the different bio-synthesis pathways of aroma-related metabolites produced by LAB and/or yeast strains.

## 4. Conclusions

In the study, the effects of different brine (fresh/aged brine) and large-scale containers (pool/jar) on the taste and aroma formation of fermented chili peppers during natural fermentation were investigated. Considering the large datasets, clustered heatmap, PLS-DA modeling, and kinetic modeling were conducted to extract the key flavor compounds distinguishing chili peppers fermented using different procedures.

As for taste properties, the results showed the effect of brine on organic acid and sugar profile was more dominant than that of containers, while FAAs content production was more affected by containers than brines. Specifically, chili peppers fermented using aged brine exhibited a significantly lower pH value and higher organic acid content than that fermented using fresh brine (*p* < 0.05). Sugar was consumed faster in chili peppers fermented by fresh brine than that by aged brine. Compared with the pickle jar, chili peppers fermented using the large-scale pool showed higher FAAs content, particularly for samples with aged brine, such as Asp, Thr, Ser, and Ala.

As for aroma, more aromatic compounds were detected in chili peppers with aged brines. Among the samples, the PAB samples showed the highest numbers of volatile compounds (96), followed by the JAB (88), PFB (80), and JFB (76). These volatile compounds mainly include esters, terpenes, alkanes, alcohols, ketones, and acids. The esters, alkanes, alcohols, acids, and ketones of aged brine were significantly higher than those of fresh brine-fermented chili peppers (*p* < 0.05), including 4-methylpentyl isobutyrate, heptyl hexanoate, amyl 2-methylbutyrate, isopentyl isovalerate, (Z)-3-hexenyl 3-methylbutanoate, heptadecane, hexadecane, octadecane, naphthalene, isohexanol, hexanol, acetic acid, 3,7-dimethyl-6-octenoic acid, and 3,5-dimethyl-2-octanone. Compared with the pickle jar, the content of terpenes in chili peppers fermented in the pool was higher, including *β*-chamigrene, (Z)-*β*-ocimene, *o*-cymene, methyl-1-naphthalene, and durene. In general, jar fermentation with aged brine contributed more flavor to pickled chili peppers than other procedures.

## Figures and Tables

**Figure 1 foods-12-00101-f001:**
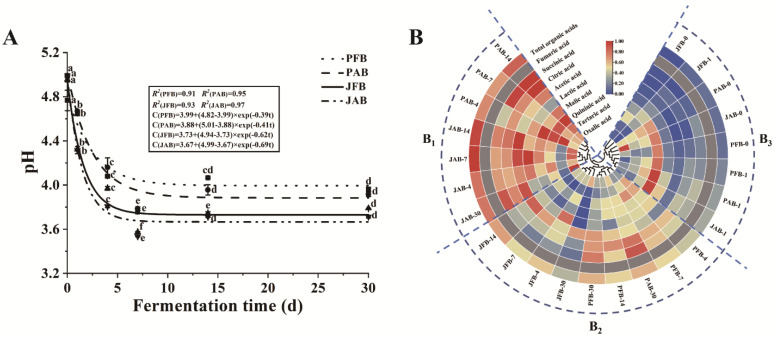
Changes in the pH value (**A**) and organic acids content (**B**) of pickled chili pepper based on different fermentation procedures, including PFB, PAB, JFB, and JAB (*n* = 3). (**A**) pH value plot: the full lines represent the fitted values by kinetic modelling and the symbols represent the experimental data. Estimated parameters of the kinetic model were shown in the box. Different lowercase letters (a–f) indicated significant differences as a function of fermentation time (*p* < 0.05). (**B**) Hierarchical clustering and heatmap visualization of organic acids: the color intensity was based on a normalized scale from a maximum of 1 (red) to a minimum of 0 (blue), which indicated the abundance of the organic acids among high (B_1_), medium (B_2_), and low (B_3_).

**Figure 2 foods-12-00101-f002:**
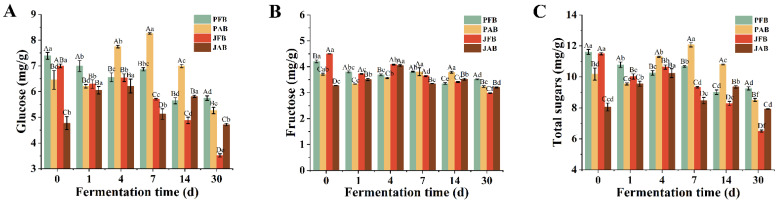
Glucose (**A**), fructose (**B**), and total sugars (**C**) of pickled chili pepper processed by PFB, PAB, JFB, and JAB (*n* = 3). Different capital letters (A–D) indicate significant differences in different treatments (*p* < 0.05); different lowercase letters (a–f) indicate significant differences as a function of fermentation time (*p* < 0.05).

**Figure 3 foods-12-00101-f003:**
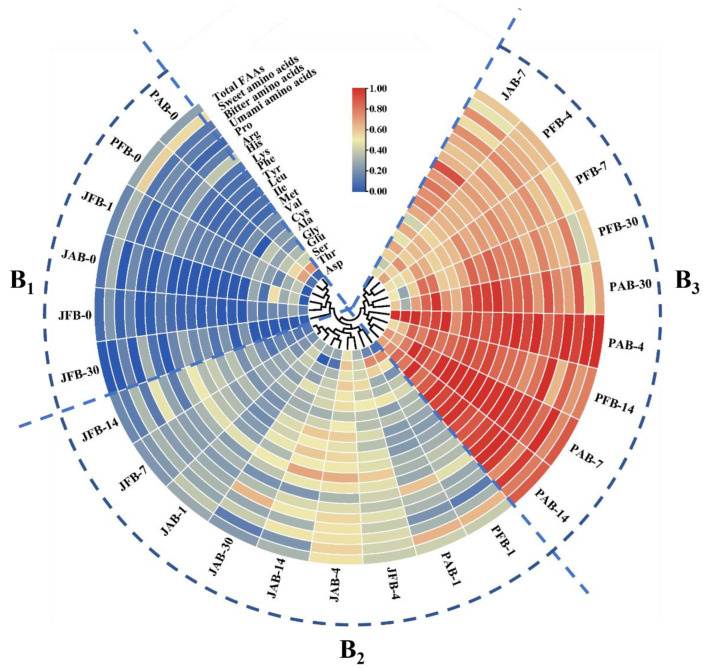
Heatmap visualization of free amino acids of pickled chili pepper processed by PFB, PAB, JFB, and JAB (*n* = 3). The color intensity was based on a normalized scale from a maximum of 1 (red) to a minimum of 0 (blue), which indicated the abundance of the free amino acids among high (B_3_), medium (B_2_), and low (B_1_).

**Figure 4 foods-12-00101-f004:**
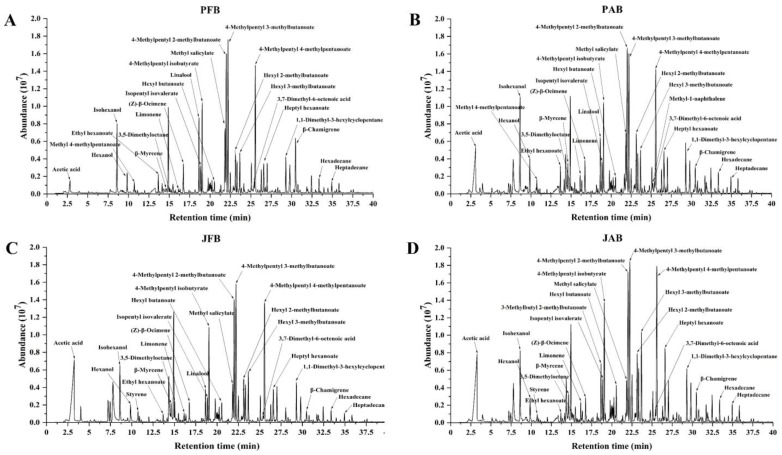
Representative total ion chromatograms of the headspace volatile compounds pickled chili pepper processed by PFB (**A**), PAB (**B**), JFB (**C**), and JAB (**D**) fermented for 30 days. The most abundant peaks were identified and marked on the chromatograms.

**Figure 5 foods-12-00101-f005:**
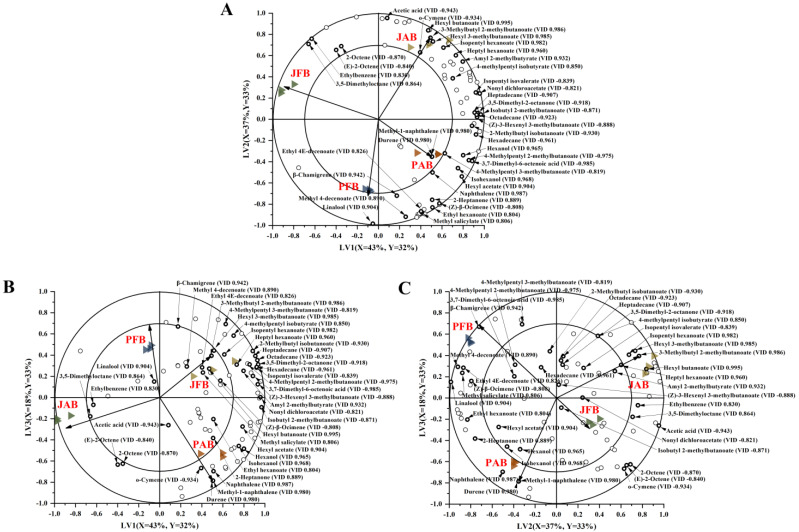
PLS−DA bi−plots of volatile compounds in the pool with fresh brine (

), pool with aged brine (

), jar with fresh brine (

), and jar with aged brine (

) for pickled chili peppers of different fermentation procedures (*n* = 3). The *X*− and *Y*−variance explained by LV1 and LV2 (**A**), LV1 and LV3 (**B**), LV2 and LV3 (**C**) were indicated in the respective axes. The open circles show different volatiles. The discriminant volatiles with absolute VID values above 0.800 were named and marked in bold.

**Figure 6 foods-12-00101-f006:**
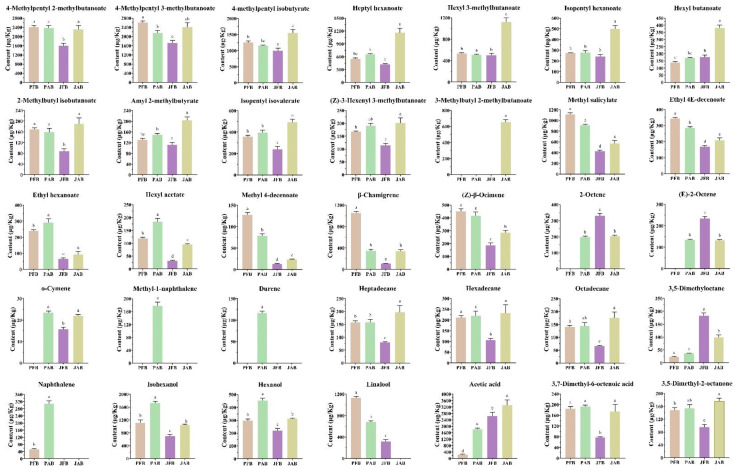
Discriminative volatile compounds content in pickled chili pepper processed by PFB, PAB, JFB, and JAB. Significant differences (*n* = 3, *p* < 0.05) were indicated with different letters.

## Data Availability

Data is contained within the article.

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
