# Peer review of "Effects of Brines and Containers on Flavor Production of Chinese Pickled Chili Pepper (Capsicum frutescens L.) during Natural Fermentation"

_foods, 2022, doi:10.3390/foods12010101_

Round 1

Reviewer 1 Report

The manuscript entitled "Effects of brines and containers on flavor production of Chinese pickled chili pepper (Capsicum frutescens L.) during natural fermentation"  covers contents of flavor compounds in pickled chili pepper depending on the freshness of the brine and type of containers. The manuscript is  well structured,  clearly described and the results are carefully discussed. The authors have done a very arduous  work on the trials and the results they have obtained.

However, a number of suggestions and comments are made on manuscript (see attached PDF file) which may be useful to increase the quality of the manuscript.

Reviewer 2 Report

The paper is generally well written. It is interesting and includes original results.

Please add the numerical data to the Abstract.

1. Introduction

The novelty of the research should be clearly indicated on the background of available literature.

2. Materials and Methods

More detailed information on the raw material should be added.

line 94: What was the sample size?

line 99: Please correct 'Ye et al. (2020)'.

2.4. Organic acids: Please add a short description of the measurement.

3. Results and discussion

The quality of some Figures should be better.
